# Age and Growth of the Threatened Smalleye Round Ray, *Urotrygon microphthalmum*, Delsman, 1941, from Northeastern Brazil

Jones Santander-Neto [1,*], Francisco Marcante Santana [2,3], Jonas Eloi Vasconcelos-Filho [3] and Rosângela Lessa [3]

1    Instituto Federal de Educação, Ciência e Tecnologia do Espírito Santo—Campus Piúma, Rua Augusto Costa de Oliveira, 660, Praia Doce, Piúma CEP 29285-000, ES, Brazil
2    Laboratório de Dinâmica de Populações Aquáticas (DAQUA), Unidade Acadêmica de Serra Talhada (UAST), Universidade Federal Rural de Pernambuco (UFRPE), Serra Talhada CEP 56909-535, PE, Brazil
3    Laboratório de Dinâmica de Populações Marinhas (DIMAR), Departamento de Pesca e Aquicultura (DEPAq), Universidade Federal Rural de Pernambuco (UFRPE), Rua Dom Manoel de Medeiros, s/n, Dois Irmãos, Recife CEP 52171-900, PE, Brazil
*    Correspondence: jones.santander@ifes.edu.br

**Abstract:** The age and growth of *Urotrygon microphthalmum* were studied using specimens captured between March 2010 and March 2012 as by-catch in the shrimp trawl fishery off the coast of the state of Pernambuco, Brazil. A total of 347 vertebrae were read, 161 from males (81.6–249.55 mm) and 186 from females (86.15–298.1 mm). The estimated average percentage index (IAPE) ranged from 0.71% to 4.33% (mean = 2.5%) in vertebrae from specimens with 1 and 6 band pairs, respectively. In the present study, the different approaches to validation produced variable results (partially valid growth zones). We then decided to discuss the growth of the species considering the formation of an annual ring. There were statistically significant differences in growth between the sexes. The best model to describe male growth was the von Bertalanffy growth model for two phases (VBGM TP) with growth parameters $L_\infty$ (maximum theoretical length) = 230.35 mm, k (growth constant) = 1.00, $t_0$ (theoretical age of size zero) = −0.76 years and for females it was the von Bertalanffy with birth size (VBGM $L_0$) model with parameters $L_\infty$ = 282.55 mm, k = 0.37. The age of maturity for males and females was 1.52 and 2.02 years, respectively, and the maximum age observed was 5.5 and 8.5 years, respectively. Despite being a fast-growing species, *Urotrygon microphthalmum* is threatened, probably due to the high mortality levels from shrimp trawling in a very narrow range of the shelf where all the life stages are captured.

**Keywords:** elasmobranch; myliobatiformes; bycatch; longevity

**Key Contribution:** First age and growth parameters for the species.

## 1. Introduction

The diversity of species of rays is greater than sharks [1,2]. However, this greater species richness does not reflect in the number of studies on age and growth available, nor on other aspects of biology. Additionally, among batoids, rays of Order Myliobatiformes (among them, the family Urotrygonidae) have been less studied than the Order Rajiformes [3,4].

The Smalleye round ray, *Urotrygon microphthalmum* occurs in shallow coastal waters of the tropical Western Atlantic Ocean at depths up to about 50 m. It is small in size and reaches 30 cm in total length [5]. Its occurrence has been recorded from Venezuela to Brazil (between the states of Amapá and Pernambuco) [6–12]. The species is characterized by matrotrophic viviparity with embryo nutrition through the yolk sac in the early stages of development and lipid histotrophy. Its reproduction is marked by low fecundity, high birth

size, rapid embryonic development, short gestation period and biannual asynchronous reproductive cycle [13]. It is classified as a second-order consumer, feeding predominantly on decapod crustaceans, especially shrimp [14].

Rays of the families Urolophidae and Urotrygonidae, (including *U. microphthalmum*) are commonly caught as bycatch in shrimp trawl fisheries [11,15–18]. This fishery captures a large quantity and diversity of bycatch fauna, in different stages of the life cycle, due to the low selectivity of the fishing gear used [19]. On the Pernambuco coast, this fishery is aimed at capturing white shrimp (*Litopenaeus schmitti*), seabob shrimp (*Xiphopenaeus kroyeri*) and pink shrimp (*Farfantepenaeus subtilis* and *F. brasiliensis*). Unlike target species, which are generally well studied, species captured as bycatch fauna are also impacted by these fisheries, but the aspects of life histories necessary for proper fisheries management are generally scarcer, which can lead to population decline.

Age information allows estimates of the growth and other vital rates such as natural mortality and longevity, which are essential for the assessment of fisheries resources and sustainable management [20]. Knowledge of age and growth parameters allows the construction of age-based population models and, together with other aspects of life history and fishing removal rates, can lead to the assessment of the population status of a species [21]. Information on the growth of species from the families Urotrygonidae and Urolophidae is recent [4,22–26] and suggests that they present rapid growth compared to species of the Rajidae family [27] or even elasmobranchs in general [28].

There is no information in the literature on the abundance of *U. microphthalmum* in fisheries and, considering its Vulnerable (VU) status according to the Brazilian Ordinance [29] and Critically Endangered (CR) according to the International Union for Conservation of Nature—IUCN [30], knowledge of biological aspects such as reproduction and growth is essential for a correct assessment of their population status. Thus, this study aims to estimate the growth parameters of *U. microphthalmum* through a multi-model inference, as well as to infer the maximum observed age and estimate the longevity and age of maturity of the species.

## 2. Materials and Methods

The analyzed specimens of *Urotrygon microphthalmum* were caught between March 2010 and March 2012 as by-catch of prawn-trawl operations. Care and use laws for experimental animals' welfare were not applied in this study due to the nature of data collection from commercial fishing landings. Some of the specimens used in this study were deposited under the voucher number LBP 0255 (Laboratório de Biologia e Genética de Peixes, Instituto de Biociências, Universidade Estadual Paulista "Júlio de Mesquita Filho", Botucatu).

The fishing gear used was twin bottom trawls. Each net was 10 m in length, 6 m at the mouth, and was formed by a 20 mm mesh in the body of the net and a 15 mm mesh in the bag. During operations, the mean velocity of trawls was 3.7 km·h$^{-1}$ (2 knots) which lasted 4 h on average. Fisheries operations targeting shrimps occurred off the coast of Pernambuco, northeastern Brazil (08°11′43″ S 34°54′13″ W and 08°38′44″ S 35°01′24″ W).

The coastal region of Pernambuco is characterized by a narrow continental platform that is relatively flat. In the area where the fleet that captures the species operates, the bottom is composed of mud and sand, and calcareous algae [31,32].

The sex and total length (*TL*, mm) of each individual were recorded, and a block containing five vertebrae was removed from the vertebral column in the thoracic portion of the coelomic cavity through the abdominal region. In the laboratory, after removing excess tissue, the vertebrae were fixed in 4% formaldehyde for 24 h and preserved in 70% ethanol [33]. Subsequently, one of the vertebrae was embedded in transparent polyester resin and then cut on a low-speed metallographic saw with a diamond cutting disc [20]. In each vertebra, a longitudinal cut was performed [34] to obtain a cut with an approximate thickness of 0.3 mm, passing through the focus [3]. Each section was mounted on a glass slide for microscopy with thermoplastic glue and polished for better visualization of the growth bands.

Following the methodology proposed by Cailliet et al. [35], two types of growth bands were considered, those being a wider opaque band and a thinner translucent band [36]. Each pair of bands was considered a ring. The birthmark was considered as a band from the change in the angle of the *corpus calcareum* [3] and was visible on all specimens (Supplementary Materials Figure S1).

The count of pairs of bands was performed under a stereoscopic microscope, with a $5\times$ magnification in a $10\times$ magnification eyepiece, using reflected light. With the aid of a micrometric ocular, the pairs of bands were counted, and the distances from the focus of each translucent band, as well as the distance from the focus to the edge of the vertebra, were measured. To assess whether the increase in the radius of the structure is proportional to the increase in size and, therefore, it is appropriate to develop the study with this rigid structure, relationships between the vertebral radius and total length were correlated in linear regression for separate sexes and then compared using ANCOVA ($\alpha = 0.05$) [37].

Two independent readings were performed without prior knowledge of the TL and the number of rings for each individual was estimated in previous readings. The following methods were used to assess precision and error between readings: the percentage of agreement between readings (PA = the number of agreements/number of vertebrae read x100) [38]; the graph of error by age of the number of bands counted in the first reading vs bands counted in the second reading and the average percentage error index (IAPE) [39] was calculated as follows:

$$\text{IAPE} = \frac{1}{N} \sum_{j=1}^{n} \left( \frac{1}{R_j} \sum_{i=1}^{n} \left( \frac{|X_{ij} - X_j|}{X_j} \right) \right) 100 \qquad (1)$$

where $N$ = Number of vertebrae; $R_j$ = the number of readings for individual $j$; $X_{ij}$ = Number of bands counted $i$ of individual $j$; $X_j$ = Mean bands counted calculated for individual $j$.

To assess the periodicity of the band pairs formation of age groups, the analysis of the marginal increment ratio (MIR) [40] was used to estimate the period in which a new ring begins to be formed through the equation:

$$\text{MIR} = \frac{VR - R_n}{R_n - R_{(n-1)}} \qquad (2)$$

where, $VR$ = Vertebral radius; $R_n$ = Radius of the last band pair formed; $R_{(n-1)}$ = Radius of the penultimate band pair formed.

Significant differences between months in MIR were analyzed using the Kruskal-Wallis test with a significance level of 0.05 and when differences were found, Dunn's post-hoc was used.

In addition, to also evaluate the periodicity of ring formation, the technique proposed by Okamura and Semba [41] was used, where the von Mises distribution model for circular data is adjusted to the frequency data of opaque and translucent bands. Three models were tested: the absence of a cycle or no pattern of mark formation (0-peak); the formation of one growth band per year (1-peak) and, finally, the model with the formation of two growth bands per year (2-peaks). Akaike's information criterion (AIC) [42] was used to assess which model best fits the data.

The following models were fitted to the observed length and age data: von Bertalanffy (VBGM) [43]; modified von Bertalanffy that fixes the beginning of the curve to the size at birth ($L_0$, in cm) according to the estimate for the species (105 mm) [13] (VBGM-$L_0$) [44,45]; von Bertalanffy for two phases (VBGMtp) [46]; Gompertz [47]; and Logistic [48] (Table 1).

**Table 1.** Growth models adjusted for length and age.

| Model | Equation |
|-------|----------|
| VBGM | $L_t = L_\infty(1 - e^{-k(t-t_0)})$ |
| VBGM $L_0$ | $L_t = L_0 + (L_\infty - L_0)\left[1 - e^{(-k)t}\right]$ |
| VBGM TP | $L_t = L_\infty\left[1 - e^{-kA_t(t-t_0)}\right]\ A_t = 1 - \dfrac{h}{(t-t_h)^2+1}$ |
| Gompertz | $L_t = L_\infty e^{[-ae(-kt)]}$ |
| Logistic | $L_t = L_\infty(1 + e^{-k(t-t_0)})^{-1}$ |

Where, $L_t$ is the length at age $t$; $L_\infty$ is the maximum theoretical length that the individual can reach; $k$ is the growth constant; $t$ is the individual's age; $t_0$ is the theoretical age of size zero; the regression parameter $A_t$ is a factor that modifies $k$ as age increases; this is the age at which the transition between the two phases occurs and $h$ determines the magnitude of the maximum age-length difference between VBGM and VBGM TP at point $t_h$.

Age-Length data were initially adjusted for grouped sexes considering reproductive seasonality [26,49]. In the study region, *U. microphthalmum* has two birth peaks, one in February and the other extending from June to October [13]. For the period of formation of the first ring, the average between the age of formation of the first band of individuals born in August (average month between June and October) and February was considered. Thus, the formation of the first band occurs between 3 and 9 months, respectively, in the case of annual band formation and, between 3 and 6 months in the case of the formation of two annual bands. In this way, considering the overall range for first band formation after birthmark (between 3 to 9 months), the age of formation of the first band of 0.5 years was defined for the population because there is no way of knowing how long it took for each neonate to form the first ring, given that the marginal increment can vary between individuals. The following growth bands follow the scenarios of the formation of one or two annual growth bands.

The parameter estimates for all models were obtained using the Excel Solver function, which uses likelihood. The likelihood tool and the bootstrap iteration function of the PopTools program [50] were used to generate confidence intervals for each parameter, based on the minimum likelihood. The method based on the minimum likelihood that uses the chi-square distribution was used for comparisons of VBGM growth parameters between sexes, as proposed by Kimura [51].

The length and age data were fitted to the growth models considering two scenarios: (s1) the formation of an annual ring and (s2) the formation of two annual rings. The results of the models were evaluated according to the Akaike Information Criterion (AIC) [42], according to the equation:

$$AIC = -2log(\theta) + 2K \tag{3}$$

where, $\theta$ = Minimum likelihood; $K$ = Number of model parameters more the error.

The difference between the *AIC* values ($\Delta_i = AIC_i - AIC_{min}$) of each model was estimated. The criterion to evaluate the statistical support of each model is described by Burnham & Anderson [52], where $\Delta_i > 10$ the model has no statistical support and can be omitted; $\Delta_i < 2$, the model has substantial support; $4 < \Delta_i < 7$, the has considerably less support. The Akaike weight ($w_i$) was used to quantify each model with respect to data fit.

$$wi = \frac{e^{(-0.5\Delta i)}}{\sum_{i=1}^{n} e^{(-0.5\Delta i)}} \tag{4}$$

Using the inverted von Bertalanffy growth curve [53], the age at maturity was estimated for maturity sizes of 187.74 and 198.73 mm estimated for males and females,

respectively [13]. Longevity ($t_x$) was estimated using the formula proposed by Cailliet et al. [35] for elasmobranchs:

$$t_x = \frac{1}{k} \ln[(L_\infty - L_0)/(L_\infty(1 - 0.95))] \tag{5}$$

where, $t_x$ = Time in which the species reaches the fraction $x$ of $L_\infty$; $k$ = Growth coefficient; $L_0$ = Birth size; $L_\infty$ = Maximum theoretical length.

## 3. Results

A total of 360 specimens (167 males and 197 females) were collected for the *U. microphthalmum* growth study. Among the vertebrae used, it was possible to read 347 (96.39%) vertebrae, 161 of which were males and 186 were females. Of these, the total length of males varied between 81.6 and 249.5 mm *TL* and of females between 86.1 and 298.1 mm *TL* (Figure 1).

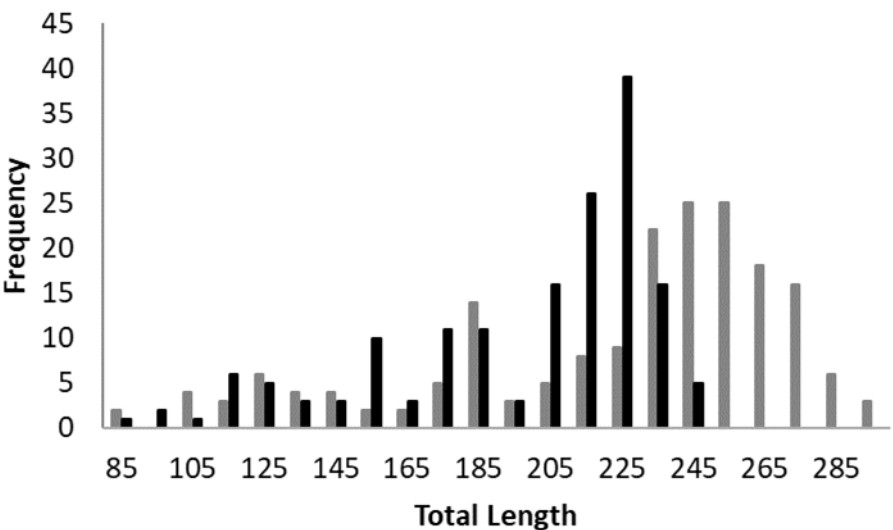

**Figure 1.** Frequency distribution of total length of *Urotrygon microphthalmum* captured in northeastern Brazil. Black bars, males; gray bars, females.

Vertebral radius (*VR*) varied between 0.34 to 1.7 mm for females, with 1 to 9 pairs of bands counted. For males, the *VR* ranged from 0.38 to 1.14 mm and 1 to 6 pairs of bands were counted.

In the relations between the vertebral radius (*VR*) and the total length (*TL*) for males and females, significant differences were observed between the sexes ($F = 1367.370$, *d.f.* = 2, $p < 0.001$). Thus, significant nonlinear relationships between *VR* and TL were estimated for males (*TL* = 141.77 × ln(*VR*) + 227.44, $r^2 = 0.894$) and females (*TL* = 133.91 × ln(*VR*) + 223, 54, $r^2 = 0.929$), indicating that the vertebrae are suitable structures for age determination as they increase proportionally with size.

The average error percentage index (IAPE) calculated was 2.51% and the variation across classes was 0.71% for the first growth band and 4.33% for the sixth band. The percentage of agreement (PA) between the readings was 74.8%, with the values of PA ± 1 and PA ± 2 equal to 98% and 99.6%, indicating that the disagreement between the readings is small. It was possible to verify that in the younger age classes the agreement between the readings is greater (number of bands < 4) than in the older age classes (Figure 2), indicating a greater variation, but even so, together with the analysis presented, they indicate a high level of reproducibility between the two readings.

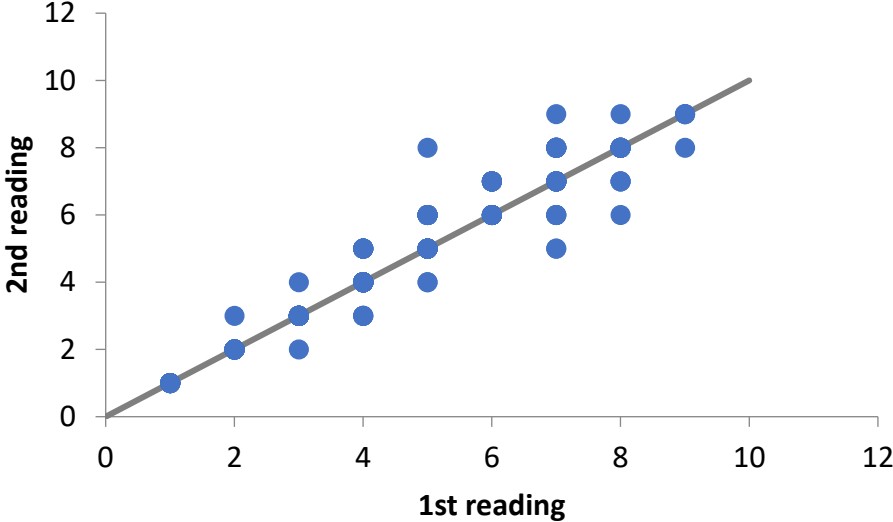

**Figure 2.** Error in the number of band pairs counted from *Urotrygon microphthalmum* captured in northeastern Brazil. The gray diagonal line indicates a one-to-one relationship.

The monthly analysis of the marginal increment ratio (MIR) showed significant differences among the months considering grouped sexes (Supplementary Materials Table S1), indicating two peaks in January and September/October followed by a fall in the IMR value (Figure 3). Considering separate genders, it was possible to observe a pattern of two peaks in MIR throughout the year for both sexes (Supplementary Materials Figure S2). When testing IMR for different age groups the result was inconclusive (Supplementary Materials Figures S3–S5).

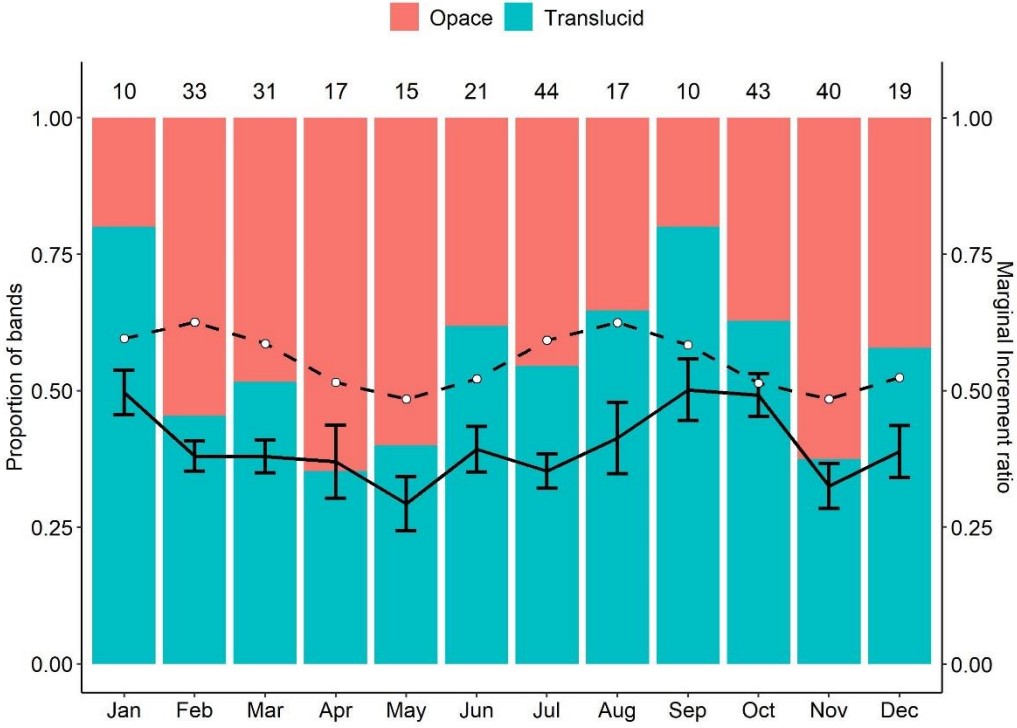

**Figure 3.** The opaque and translucent proportion of vertebral band (dashed line) and marginal increment ratio (continuous line with standard deviation) of *Urotrygon microphthalmum* from northeastern Brazil.

In the analysis of opaque and translucent bands (Figure 3), the hypothesis of the absence of band formation pattern had the lowest *AIC* value (478.19), followed by the formation of two bands per year (*AIC* = 478.48; $\Delta_i$ = 0.29) with substantial support and one band per year (*AIC* = 481.88; $\Delta_i$ = 3.69) with less data support. None of the models was disregarded in the analysis ($\Delta_i$ >10).

Initially, the grouped sexes data were fitted to the growth models considering the two scenarios. The lowest *AIC* value and highest Akaike weight ($w_i$) were estimated for the VBGM considering an annual ring (Supplementary Material Table S2). From this, the comparison between sexes indicated significant differences in growth between males and females ($\chi2$ = 11.17; *g.l.* = 3; *p* = 0.011).

The female and male data were then fitted separately to the growth models considering the two ring formation hypotheses. For females, there was a tie in the lowest *AIC* value with VBGM-$L_0$ presenting the same value for both hypotheses (Table 2; Figure 4). All models showed high to moderate statistical support considering the *AIC*. The von Bertalanffy models and derivatives presented very similar growth parameters within each scenario. Considering the $L_\infty$, the estimated values for these models fit within the range of total lengths used in the sample. For males, the VBGM-TP considering the formation of a ring presented the lowest *AIC* value (Table 2). For this scenario, all models showed high to moderate statistical support. However, considering the $L_\infty$, the estimated values (considering the confidence intervals) for the VBGM-TP are well below the range of total lengths of individuals from which vertebral were collected. The other von Bertalanffy models in this scenario presented $L_\infty$ values that fit within the range of total lengths sampled and growth constants similar to each other, differing significantly from the overestimated values of the VBGM-TP. For males in the scenario of the formation of two annual rings, the VBGM-TP was also the model with the lowest *AIC*, and, in this case, several models did not show statistical support.

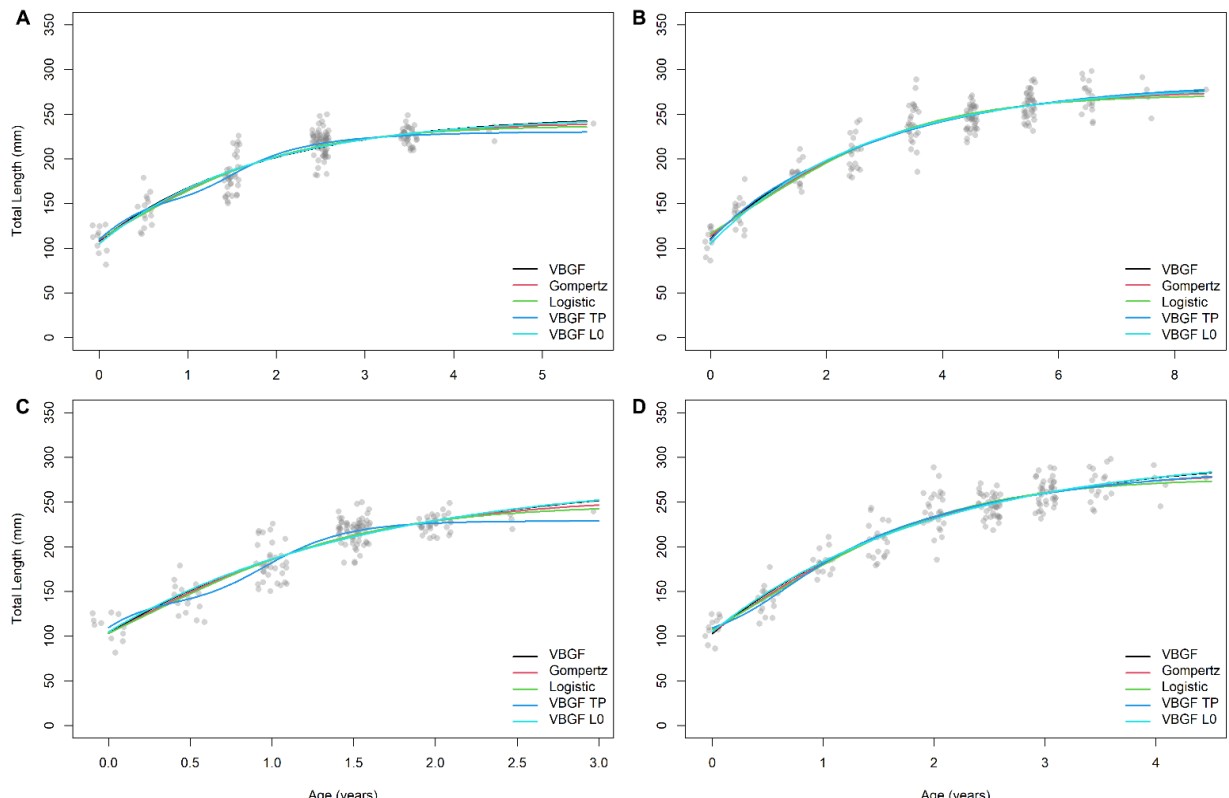

**Figure 4.** Growth curves estimated for *Urotrygon microphthalmum* from northeastern Brazil. (**A**) male, one ring per year; (**B**) female, one ring per year; (**C**) male, two rings per year; (**D**) female, two rings per year. Gray circles are observed age.

**Table 2.** Comparison between growth models for grouped sexes ranked based on the Akaike Information Criterion (*AIC*); $\Delta_i$ = Akaike difference; $w_i$ = Akaike's weight. *K* = the number of model parameters. $L_\infty$ = theoretical maximum length; *k* = growth constant; $t_0$ = theoretical age where the length of the fish is zero; $L_0$ = Estimated length at age 0. The values between parenthesis are the 95% confidence intervals of parameters.

| Dataset | Model | $L_\infty$ | *k* | $t_0$ | *h* | $t_h$ | $AIC_i$ | $\Delta_i$ | $w_i$ |
|---|---|---|---|---|---|---|---|---|---|
| Female—1 ring | **VBGM-L$_0$** | **282.55 (272.63/292.47)** | **0.37 (0.32/0.42)** | | | | **1575.68** | **0.00** | **0.28** |
| | Gompertz | 277.03 (268.11/285.95) | | | | | 1577.45 | 1.76 | 0.12 |
| | VBGM-TP | 286.74 (270.55/302.93) | 0.34 (0.24/0.45) | −1.31 (−2.06/−0.57) | −0.06 (−0.36/0.24) | 0.55 (−2.82/3.92) | 1579.70 | 4.01 | 0.04 |
| | Logistic | 271.70 (264.47/278.94) | 0.62 (0.53/0.70 | 0.47 (0.33/0.61) | | | 1580.58 | 4.90 | 0.02 |
| | VBGM | 286.01 (273.79/298.23) | 0.35 (0.28/0.41) | −1.41 (−1.70/−1.13) | | | 1582.61 | 6.93 | 0.01 |
| Female—2 rings | **VBGM-L$_0$** | **304.59 (287.78/321.39)** | **0.50 (0.41/0.58)** | | | | **1575.68** | **0.00** | **0.28** |
| | Logistic | 276.62 (268.23/285.04) | 1.07 (0.93/1.22) | 0.43 (0.34/0.51) | | | 1577.17 | 1.48 | 0.13 |
| | Gompertz | 285.54 (274.10/296.60) | | | | | 1578.31 | 2.63 | 0.08 |
| | VBGM-TP | 294.08 (254.41/333.73) | 0.50 (0.01/1.00) | −1.42 (−3.99/1.14) | 0.37 (−0.20/0.94) | 0.18 (−0.50/0.86) | 1579.93 | 4.24 | 0.03 |
| | VBGM | 302.45 (284.42/320.47) | 0.51 (0.41/0.62) | −0.80 (−0.98/−0.62) | | | 1582.61 | 6.93 | 0.01 |
| Male—1 ring | **VBGM-TP** | **230.35 (220.48/240.21)** | **1.00 (0.56/1.44)** | **−0.76 (−1.03/−0.48)** | **0.34 (0.11/0.56)** | **1.16 (0.86/1.47)** | **1343.66** | **0.00** | **0.33** |
| | Logistic | 237.33 (228.59/246.06) | 0.95 (0.79/1.11) | 0.14 (0.02/0.26) | | | 1345.15 | 1.49 | 0.15 |
| | Gompertz | 241.87 (231.05/252.68) | | | | | 1346.60 | 2.94 | 0.07 |
| | VBGM-L$_0$ | 247.42 (234.85/259.99) | 0.57 (0.46/0.68) | | | | 1347.25 | 3.59 | 0.05 |
| | VBGM | 249.18 (234.54/263.82) | 0.55 (0.41/0.68) | −1.04 (−1.30/−0.77) | | | 1348.81 | 5.15 | 0.02 |
| Male—2 rings | VBGM-TP | 288.85 (220.33/237.38) | 2.33 (0.95/3.70) | −0.47 (−0.69/−0.24) | 0.59 (0.42/0.76) | 0.68 (0.52/0.84) | 1343.78 | 0.12 | 0.31 |
| | VBGM-L$_0$ | 278.19 (249.98/306.40) | 0.63 (0.46/0.81) | | | | 1347.25 | 3.59 | 0.05 |
| | Logistic | 246.82 (234.42/259.22) | 1.44 (1.18/1.70) | 0.22 (0.14/0.31) | | | 1352.19 | 8.54 | 0.00 |
| | Gompertz | 256.86 (239.42/274.31) | | | | | 1356.78 | 13.12 | 0.00 |
| | VBGM | 275.87 (246.37/305.36) | 0.65 (0.44/0.87) | −0.71 (−0.91/−0.51) | | | 1362.23 | 18.58 | 0.00 |

Considering the VBGM, the estimated age of maturity for males was 1.51 and 1.04 years for one and two rings, respectively. The estimated age at maturity for females was 1.98 and 1.24 years for one and two rings, respectively. The sample is composed of adult individuals (Figure 5), with males being more abundant in the 2.5-year class and females in the 4.5 and 5.5-year classes.

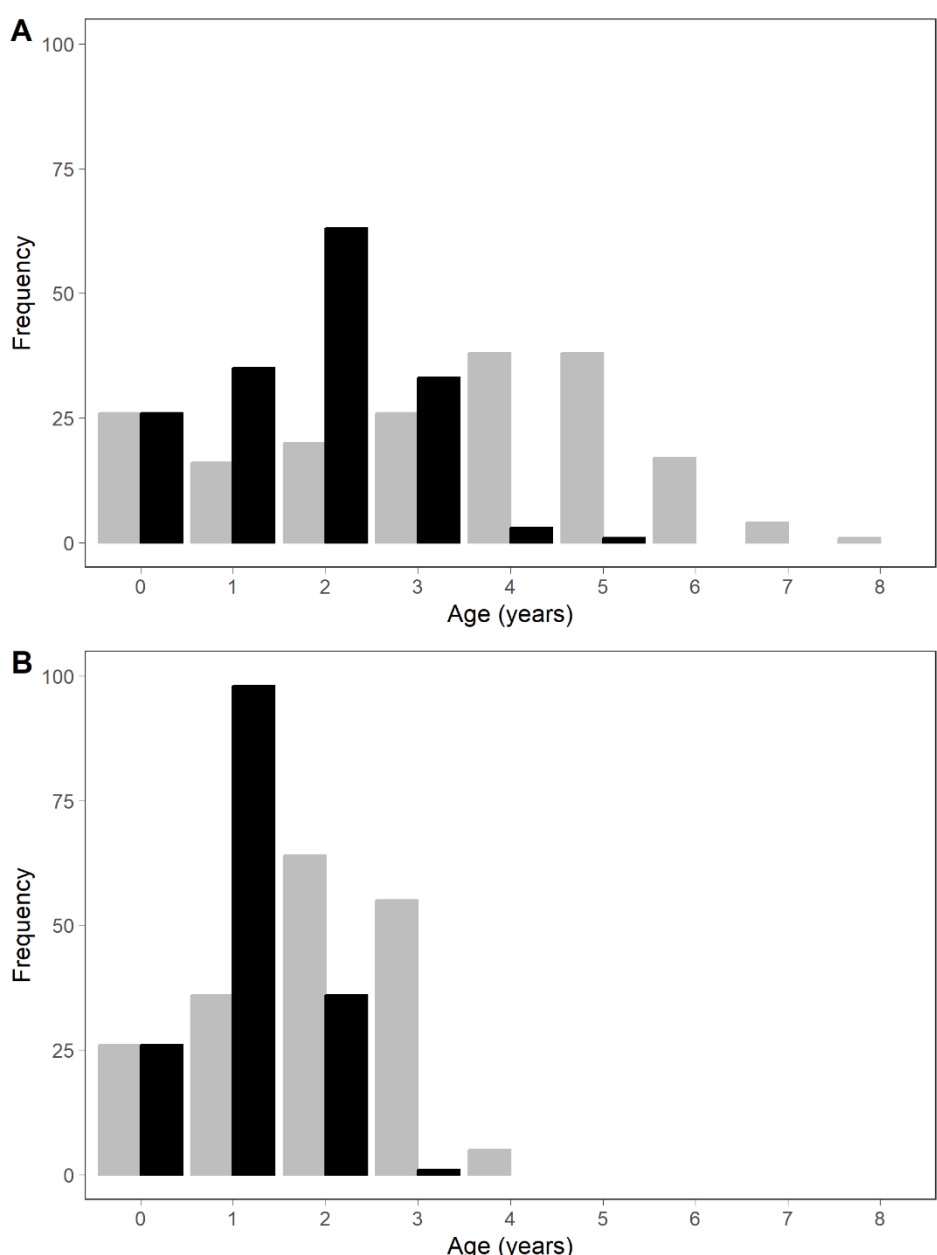

**Figure 5.** Age distribution for *Urotrygon microphthalmum* from northeastern Brazil, considering (**A**) one ring per year and, (**B**) two rings per year. Black bars, males; gray bars, females.

The estimated age at which the species reaches 95% of $L_\infty$ (longevity) for males was 4.45 and 3.87 years for one and two rings, respectively. For females, the estimated longevity was 7.25 and 5.04 years for one and two rings, respectively. Through direct observation of ages considering a ring, the oldest male was 5.5 years old, and the oldest female was 8.5 years old. Considering two rings, the oldest male was 3 years old, and the oldest female was 4.5 years old (Figure 5).

## 4. Discussion

This work is the first age and growth study of the critically endangered *Urotrygon microphthalmum* and the first for the family in the Atlantic. The results obtained were compared with those obtained for *Urotrygon aspidura*, *U. rogersi* and *Urobatis halleri* (Urotrygonidae) in the Pacific and, among the species *Urolophus lobatus*, *Urolophus paucimaculatus*, *Trygonoptera personata* and *T. mucosa* in the Indian Ocean, due to their greater phylogenetic proximity, based on morphological characters, with these species [54]. Our results are similar to those found for the rays of this group regarding the indication that they are fast-growing and short-lived species, which may have important implications for the evaluation of the population status and management of this species since it occurs in very narrow portions of the continental shelf under the strong influence of trawling.

During this study, vertebrae of 347 individuals were used for the age and growth study of *U. microphthalmum* and, according to Thorson & Simpfendorfer [55], this sample size is sufficient ($n > 200$) for good precision in estimating the growth parameters, if we consider that the sample had a reasonable proportion of males and females and all length classes were represented. Age and growth studies in elasmobranchs generally use samples from fisheries targeting commercially exploited species. This characteristic can lead to a sample that is not very representative of the population or even to the absence of certain length classes. However, despite being by-catch fauna, the capture of the stingray *Urotrygon microphthalmum* covered all length classes from neonates to large adults close to the maximum size reported in works on the species [5,8,13].

The use of vertebrae for age and growth studies is the most used approach in elasmobranchs [3] because this structure has good calcification and visualization of growth bands in most species since the growth bands are not reabsorbed as in other tissues. Although the vertebrae of *U. microphthalmum* are small, the visualization of the growth bands was adequate for the study, with 96.39% ($n = 347$) of the 360 vertebrae being used for reading, with good visualization of the growth bands and similarity between readings. Vertebrae from the Urolophids *Urolophus lobatus*, *U. paucimaculatus*, *Trygonoptera personata*, *T. mucosa*, *Urobatis halleri* and the congeners *U. aspidura* and *U. rogersi* proved equally suitable for growth studies [4,22–26].

Validation of the period of band formation is considered one of the most critical steps when using rigid frameworks for age estimation [56–58], yet few studies have rigorously validated the temporal periodicity, which consists of validating all age classes, validating only some portion of the species' life history [58]. In the present study, the different approaches produced variable results, which would classify the validation result as "partially valid growth zones" according to Cailliet [58]. As already observed for species of tropical elasmobranchs, validation of the periodicity of band formation is quite difficult to visualize, being defined for some species from other studies with the same species or even following what is defined in the literature for the genus [59,60]. Additionally, in the results of parameter estimation through multi-model inference, the lowest likelihood values, that is, the best fit, were obtained considering only one ring. Considering, (1) the difficulties in validating the growth of tropical species; (2) the likelihood results for estimates with an annual ring and (3) annual depositions for the Urotrygonidae e Urolophidae species [4,21–24,26], we decided to discuss the data of the species considering the formation of an annual ring, although we are presenting the results for both hypotheses. We strongly recommend that alternative approaches be applied to validate the periodicity of band formation of *U. microphthalmum*, such as marking and injection of oxytetracycline (OTC), to overcome the difficulty of validation in the present study.

Chondrichthyes have diverse reproductive cycles and not all species have an annual seasonal cycle, with some reproducing in a partially defined annual cycle with one or more seasonal peaks to species that reproduce throughout the year [61]. The stingray *U. microphthalmum* has a biannual asynchronous reproductive cycle with two peaks of births during the year and one of them with an extended birth period for this region [13], so it is not possible to state the age of formation of the first growth band after the birthmark.

Therefore, the average age for the formation of the first band was determined based on the reproductive biology of the species to reduce errors in the growth analysis caused by reproductions lasting several months, as suggested by Harry et al. [49].

The use of different models to estimate the growth parameters of elasmobranchs has become a more frequent approach due to the need and requirement to evaluate the existence of species that present different growth patterns, allowing better estimates of the parameters in contrast to the previous choice. of a given model (usually VBGM) to describe growth [3,4,26,35,49,59,60,62–65].

For estimates of female growth parameters, the VBGM with $L_0$ had the lowest *AIC* value and the highest Akaike weight, thus being considered the most robust among the selected models. The model that showed high statistical relevance was the Gompertz model, which tends to fit better to rays of the order Myliobatiformes, as this includes species in which the volume increases much more with age than the length or width of the disk [3]. Moderately relevant models were VBGM TP, Logistic and VBGM. No model showed a $\Delta_i$ greater than 10 and should be discarded. The three von Bertalanffy models (VBGM $L_0$, VBGM TP and VBGM) presented very similar $L_\infty$ and $k$. Additionally, these models presented an $L_\infty$ closer to the maximum sample size, especially for VBGM TP and VBGM. The Gompertz and Logistic models tend to underestimate this parameter. Despite the best model being the two-parameter VBGF, this has to be carefully evaluated because it results in biased growth estimates even with light variations in the value of fixed $L_0$ [66]. If we consider the U. microphthalmum has a large variation around the estimated $L_0$ [13] its recommended the using the three-parameter VBGF as proposed by Pardo et al. [66].

For estimates of male growth parameters, the VBGM-TP had the lowest *AIC* value and the highest Akaike weight, thus being considered the most robust among the selected models. Despite this, the $L_\infty$ estimated through this model was very low considering the largest individual in the sample and the k was much higher than the values found for species of the same family [4,22–24,26]. The model that presented high statistical relevance was the Logistic one. Moderately relevant models were Gompertz, VBGM $L_0$ and VBGM. No model showed a $\Delta_i$ greater than 10 and should be discarded. Among the von Bertalanffy models (VBGM TP, VBGM L0 and VBGM), VBGM $L_0$ and VBGM presented very similar $L_\infty$ and $k$. Additionally, these models presented an $L_\infty$ closer to the maximum sample size.

The estimated growth parameters for *U. microphthalmum* differed between sexes as observed for the other species of the Urolophidae and Urotrygonidae families [22–24,26], due to differences in size, maximum age and maturation age, and thus, growth parameters are treated separately between sexes. For both sexes, the estimated $L_\infty$ was lower than the largest individual in the sample. The reason why the estimated asymptotic maximum length was lower than the observed maximum length is because the estimate refers to the maximum average length that the species would reach if it grew forever and not just the maximum length that it could reach. Asymptotic maximum length estimates lower than the sample maximum lengths were also observed for males and females of *Urolophus lobatos* and *U. paucimaculatus* [22,24] and for males of *Urotrygon rogersi* and *U. aspidura* [4,26] (Table 3). Females reach greater asymptotic lengths in disk width ($WD_\infty$) than males for the species compared, except for *Urobatis halleri* where females had lower $WD_\infty$. For this species, the growth curve did not reach the asymptote and large females were not collected due to reproductive seasonality [25], indicating that the $WD_\infty$ reported by Babel [67] (310 mm) is more faithful to the biology of the species.

**Table 3.** Growth parameters of related species of the families Urolophidae and Urotrygonidae. $DW_\infty$, asymptotic disc width (in mm); $k$, annual growth rate; $t_{max}$, the maximum age (in years); $DW_{max}$, maximum disc width (in mm); $t_{mat}$, age at maturity (in years). * $TL_\infty$ converted to $DW_\infty$; ** DW of the oldest specimens sampled.

| Species | Model | Sex | $DW_\infty$ | $k$ | $t_{max}$ | $DW_{max}$ | $t_{mat}$ | n | Study Area | Reference |
|---|---|---|---|---|---|---|---|---|---|---|
| *Urotrygon microphthalmum* | VBGM L0 | F | 147 * | 0.37 | 8.5 | 154 | 2 | 186 | SW Atlantic, Brazil | Present study |
| | VBGM TP | M | 127 * | 1.00 | 5.5 | 128 | 1.5 | 161 | | |
| *Urotrygon aspidura* | VBGM TP | F | 249 | 0.47 | 7.5 | 265 | 2.3 | 125 | E Pacific, Colômbia | [4] |
| | VBGM TP | M | 160 | 1.63 | 5.5 | 185 | - | 184 | | |
| *Urotrygon rogersi* | VBGM TP | F | 200 | 0.22 | 8 | 199 | 1 | 234 | E Pacific, Colômbia | [26] |
| | VBGM TP | M | 155 | 0.64 | 6 | 170 | 1 | 232 | | |
| *Urobatis halleri* | VBGM | F | 224 | 0.15 | 14 | 213 ** | 3.8 | 96 | NE Pacific, USA | [25] |
| | VBGM | M | 286 | 0.09 | 14 | 239 ** | 3.8 | 84 | | |
| *Urolophus lobatus* | VBGM | F | 249 | 0.37 | 15 | 277 | 3.1 | 388 | SE Indian, Australia | [22] |
| | VBGM | M | 210 | 0.51 | 13 | 237 | 1.7 | 428 | | |
| *Urolophus paucimaculatus* | VBGM | F | 261 | 0.26 | 14 | 272 | 5 | 330 | SE Indian, Australia | [24] |
| | VBGM | M | 243 | 0.36 | 11 | 256 | 3.5 | 437 | | |
| *Trygonoptera personata* | VBGM | F | 303 | 0.14 | 16 | 311 | 4 | 352 | SE Indian, Australia | [23] |
| | VBGM | M | 269 | 0.20 | 10 | 269 | 4 | 303 | | |
| *Trygonoptera mucosa* | VBGM | F | 308 | 0.24 | 17 | 369 | 5 | 324 | SE Indian, Australia | [23] |
| | VBGM | M | 261 | 0.49 | 12 | 283 | 2 | 400 | | |

Size sexual dimorphism in elasmobranchs is well documented and can be evidenced in length frequency distributions, length-weight ratio, size at maturity and age at maturity [27,28]. Females of *U. microphthalmum* reached ages and lengths greater than males, such as *U. aspidura*, *U. rogersi*, *Urolophus lobatus*, *U. paucimaculatus*, *Trygonoptera personata* and *T. mucosa* [4,22–24,26]. This age and length of sexual dimorphism in animals may reflect an adaptation to different reproductive modes [68] and the explanation for this phenomenon may be quite complex, involving several factors such as mating success, fecundity, growth and foraging success, but it seems it is clear that larger sizes consist of an evolutionary advantage [69], either allowing an increase in fecundity or larger and more capable embryos [70,71].

Males had higher $k$ values than those found for females, except for *Urobatis halleri*, probably because males reached larger sizes than females for the study. According to the authors, this is probably due to the shallower beach seines biasing towards smaller females [25]. For males, the growth constant was lower than that found for the congener *U. aspidura* and higher than that found for *U. rogersi*, as well as that found for the species of the Urolophidae family. In females, the growth constant was higher than that found for *U. rogersi* and lower for *U. aspidura*, and similar to that found for *Urolophus lobatus*. The maximum age observed for both sexes is similar to that found for *U. rogersi* and *U. aspidura*, and lower than that found for the other species of the Urolophidae family (Table 3). For batoids, the growth constant generally varies between 0.1 and 0.3 [72] indicating that both sexes of *U. microphthalmum* are beyond this upper limit, however, in general, Myliobatiformes present faster growth (0.2 to 0.5) [3]. In this way, the species has fast growth and low longevity compared to most elasmobranchs [27,28].

Estimates of growth models are affected by several factors, such as sample size, fishing gear selectivity, range of lengths, age verification methodology, validation, and growth model adjustment techniques [3,55,73] and, therefore, interspecific comparisons may be discrepant due to the bias caused by these factors. Furthermore, growth can be divided into a series of stages in the life history of a species and changes between stages are characterized by changes in the rate of development as maturity, changes in behavior or habitat [46,74]. In the present study, we could observe that the growth model that best fitted the data can be one of the sources of considerable discrepancy between the growth parameters and observed length data.

The estimated longevity of *Urotrygon microphthalmum*, following the function proposed by Cailliet et al. [35] was lower than the maximum age reported, indicating that the formula used is not adequate to estimate longevity in this species. Among the reasons we can highlight the high variation in birth size or the best-fitted models in the present study do not accurately describe the growth of the species, evidencing that the choice of the best model should not be made exclusively considering the fit of the model.

The difference in age at maturity between males and females was small (<1 year). This small difference was also observed for *U. rogersi*, *Urobatis halleri* and *Trygonoptera personata* whereas the species *Urolophus lobatus*, *U. paucimaculatus* and *Trygonoptera mucosa* showed a greater difference (>1 year) between the sexes (Table 3). Species of the Urotrygonidade family had the lowest maximum ages among the compared species, as well as the lowest maturation ages. These species mature at much lower ages compared to the average for batoids (8.6 years) [27], indicating that they are species with early sexual maturation.

From the growth pattern found in this work and associated with the reproductive parameters of the species [13], *Urotrygon microphthalmum* is a fast-growing species, with early sexual maturation, low fecundity, and short life. Considering the current scenario where 1/3 of elasmobranch species are threatened [75], even species with biological information that may suggest a less vulnerable species must have evaluated the mortality levels to which it is subjected and the relationship with the survival of the most relevant life stages, as a fundamental analysis to characterize its population status, since the species *Urotrygon microphthalmum* itself is categorized as critically endangered [30] probably due to occupying a very narrow range on the shelf that is under great pressure from fishing mortality due to shrimp trawling.

**Supplementary Materials:** The following supporting information can be downloaded at: https://www.mdpi.com/article/10.3390/fishes8030160/s1, Figure S1: Band pairs formed on the sectioned vertebrae of *Urotrygon microphthalmum*. BM, birthmark; Figure S2: IMR median between months for male and female of *Urotrygon microphthalmum*; Figure S3: IMR between months for both sexes of *Urotrygon microphthalmum*. G1: 2–4 bands; G2: 5–7 bands; G3: 8–9 bands; Figure S4: IMR between months for females of *Urotrygon microphthalmum*. G1: 2–4 bands; G2: 5–7 bands; G3: 8–9 bands; Figure S5: IMR between months for males of *Urotrygon microphthalmum*. G1: 2–4 bands; G2: 5–7; Table S1: Kruskal-Wallis test/post-hoc de Dunn from grouped sexes between months of *Urotrygon microphthalmum*; Table S2: AIC from growth models for grouped sexes of *Urotrygon microphthalmum*.

**Author Contributions:** Conceptualization, J.S.-N. and R.L.; methodology, J.S.-N., F.M.S., J.E.V.-F. and R.L.; software, J.S.-N. and J.E.V.-F.; formal analysis, J.S.-N. and J.E.V.-F.; investigation, J.S.-N.; resources, J.S.-N. and R.L.; data curation, J.S.-N.; writing—original draft preparation, J.S.-N.; writing—review and editing, J.S.-N., F.M.S., J.E.V.-F. and R.L.; supervision, R.L.; project administration, J.S.-N. All authors have read and agreed to the published version of the manuscript.

**Funding:** This research received no external funding.

**Informed Consent Statement:** Not applicable.

**Data Availability Statement:** The datasets generated and analyzed during the current study are not publicly available but are available from the corresponding author upon reasonable request.

**Acknowledgments:** R.L. thanks the Productivity Grant (PQ 310200/19) from the Conselho Nacional de Desenvolvimento Científico e Tecnológico (CNPq).

**Conflicts of Interest:** The authors declare no conflict of interest.

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
