# Peer review of "Age and Growth of the Threatened Smalleye Round Ray, Urotrygon microphthalmum, Delsman, 1941, from Northeastern Brazil"

_fishes, doi:10.3390/fishes8030160_

Round 1

Reviewer 1 Report

I have reviewed the manuscript “Age and growth of the threatened Smalleye Round Ray, Urotrygon microphthalmum, Delsman, 1941, from northeastern Brazil” and am of the opinion that it is not suitable for publication until after revision. There are some sections that require clarity and greater detail before results can be adequately scrutinized. Specific comments are included below.

Line 23: What is meant by “discuss the data of the species”?

Line 24: Were these differences statistically significant?

Line 25 and throughout: Define acronyms at first use.

Lines 25 and 26: Why is t sub 0 not reported for females?

Line 80: Change “aiming” to “targeting”.

The animal health and welfare statement comes out of nowhere and is likely better suited elsewhere.

Lines 90-91. Provide a more clear description of where vertebrae were removed from the vertebral column.

Line 97: Was there a significant amount of polishing or was it done to simply smooth rough surface from cutting?

Line 100: Insert “those” before “being”.

Line 101-102: The birthmark and associated band needs to be better described. Also, was it visible on all specimens regardless of month of parturition? 

Line 106: Delete “the structure to”.

Line 108: How were the lengths adjusted? What was the purpose of this?

Line 109: What is CT? Define.

Line 112: Does PC represent.... percent agreement?

Line 119: Is this supposed to be the periodicity of growth band formation?

Line 125: Significant differences in what?

Lines 132-134: Change to “bands” not “brands”.

Line 135: Were models “adjusted’ or were data fit to the models?

Line 144: Remove comma before At and italicize.

Line 146: Be consistent with use of abbreviations and acronyms (see line 136).

Line 155: This needs a more complete explanation. Why only adjust for six months when there is a range?  Depending on how you adjust can significantly impact your growth model parameter estimates.  

Line 157: Parameter estimates were obtained.

Line 163: Based on previous text explain bands, me clear if you are referring to a single band or single band pair.

Line 176-177: Where did length values come from? Why did you not use logistic regression to estimate size at 50% maturity as is commonly reported.

Lines 184-186: Why is this here?

Line 200: What does g.l. denote?

Line 201-203: This is not clear and should be revised. Why were the relationships “adjusted”. How do non-linear relationships indicate vertebrae are suitable for ageing?

Line 206: The acronym PC was already defined. PC +1 and +2 could be confusing. I assume you are referring to counts within 1 year and 2 years.

Lines 208-211: This needs to be revised as the meaning and intent are not clear.

Line 217: Change between to among.  Change gender to sex. Gender is a social term.

Line 220: I do not think this figure should be a supplementary figure. It is important in establishing periodicity of band formation.

Line 223-227: This is not well explained and should be revised.

Figure 3: Remove “s” from opaque. What is “bord”? What do lines represent?  What type of error bars are displayed?

Line 233: It is still not clear how or why data were grouped and how they were adjusted. For reason outlined above, change gender to sex throughout.

Line 244: What does length used in the sample mean?

Line 248: Same.

Line 278-282: If the sample was almost entirely composed of adults how was size at maturity determined?  This needs to be clearly explanained in the discussion.

Lines 281-282: Year class?

Line 296: Why is the word “basically” used?

Discussion. It would be meaningful to discuss mean brood size, size at maturity and longevity to show potential lifetime fecundity and how that could impact conservation efforts.

Line 305: Find a more accurate word than adjust. It does not work here or other laces it is used.

Line 305: This is not true. It would depend on if all size classes are represented, variability in size at age, number of females and males samples, etc.

Line 315: What does “presents good calcification” mean?

Lines 317-318: What is meant by “visualization of the growth bands was adequate for the study, with 96.39% of the 317 vertebrae being used for reading, and consequent good visualization of the growth bands.”? This needs revision for clarity.

Lines 322-325: No new paragraph at end.

Lines 326-340: Unclear if individual bands or band pairs are being discussed.

Line 355: Should include discussion of the Pardo et al 2013 paper that indicates that the three parameter VBGF is most reliable and the two parameter model should be avoided.

Line 378: No new paragraph.

Line 395: No new paragraph.

Lines 401-411: Should directly discuss some of the unlikely k values that were obtained and indicate that the growth model for males should be revisited based on this and other issues pointed out within the text. Also, speculate why there was an issue.

Line 428: Or that the growth model in the current study does not accurately describe growth. Are there other studies that show theoretical longevity is off based on longevity estimates from recaptured tagged sharks of known or approximately known age?

Lines 430-438: Clarity on how age/size at maturity was determined in the absence of immature individuals will need to be presented in the methods and results before this can be considered meaningful.  

Reviewer 2 Report

The authors study the age and growth of Urotrygon microphthalmum using specimens captured between March 2010 and March 2012 as by-catch in the shrimp trawl fishery off the coast of the state of Pernambuco, Brazil.

The sampling period seems to me too far back in time, more than 10 years have passed and the number of specimens to be analyzed has not increased, also to understand what has happened to the resource in response to the climate changes that have occurred in recent years, if there have been changes in fishing effort, number of fishing vessels and modernization of fishing gear.

Also, a description of the analyzed vertebrae, photos and more are missing.

The authors under discussion reported that they test different models on the various hypotheses, i.e. formation of 1 or 2 annual rings but then consider only the formation of one annual ring. I don't understand.

Round 2

Reviewer 1 Report

After reviewing the revised manuscript, it is clear that the authors did not address all of my concerns and were often vague in their response. I have addressed my remaining concerns below and feel the manuscript is acceptable for publication after these concerns are specifically addressed. I assume there will be editorial grammar assistance as it is need in sections.  

Line 25 and throughout: Define acronyms at first use.

Answer: Defined

Reviewer: Not done

The animal health and welfare statement comes out of nowhere and is likely better suited elsewhere.

Answer: We changed to right after first sentence of this section

Reviewer: This should be moved to the end of the manuscript perhaps in the Acknowledgements. While it was moved the statement is made on lines 77-82 and 91-95.

Lines 90-91. Provide a more clear description of where vertebrae were removed from the vertebral column.

Answer: Done as suggested

Reviewer: Were they cervical or thoracic?  Were they removed from under the first dorsal fin? If not, where. This is important as there can be variability in age estimates depending where they are taken from the vertebral column.

Line 108: How were the lengths adjusted? What was the purpose of this?

Answer: linear regression added in the text. To detect differences

Reviewer: This does not answer the question or provide clarity.  

Line 155: This needs a more complete explanation. Why only adjust for six months when there is a range? Depending on how you adjust can significantly impact your growth model parameter estimates.

Answer: We elucidate. Anyway, given the biology of the species, we will invariably have a variation between the real time of formation of the first band for each individual and the estimated time.

Reviewer: This still needs clarification. I assume the intent is to convey that 0.5 was removed from all band counts to determine age.

Line 163: Based on previous text explain bands, me clear if you are referring to a single band or single band pair.

Answer: Explained. Age data

New Comment, Line 168: Should be parameter.

Line 176-177: Where did length values come from? Why did you not use logistic regression to estimate size at 50% maturity as is commonly reported.

Answer: they came from size at 50% maturity from reference nº13. They use logistic regression to estimate L50%

Reviewer: Then state explicitly.

Line 201-203: This is not clear and should be revised. Why were the relationships “adjusted”. How do non-linear relationships indicate vertebrae are suitable for ageing?

Answer: Although it is very difficult to observe this, if the increase in size of individuals were not proportional to the increase in the vertebrae radius, this structure would not be suitable for an age and growth study

Reviewer: This does not address my question. Why would differences between the sexes and a subsequent “adjustment” provide support that vertebrae a suitable for ageing? This needs editing for accuracy and clarity.

Lines 208-211: This needs to be revised as the meaning and intent are not clear.

Answer: We are explaining that even with older individuals having less agreement between readers than younger individuals, the difference is suitable

Reviewer: This sentence still needs editing. Perhaps change the word “initial” to “younger age classes”.

New comment, Line 234: Should be “were”.

Line 233: It is still not clear how or why data were grouped and how they were adjusted. For reason outlined above, change gender to sex throughout.

Answer: Initially we do not know if the growth is different from male and female. And ir order to test this, we have to estimate parameters for grouped sexes to compare with parameters estimated from male and female to then assess if there is significant differences

Reviewer: Why were data adjusted?  What is meant by “adjustment”?  This needs to be clarified. Are the authors saying the data were fit to the model?

Line 244: What does length used in the sample mean?

Answer: total length. Added in the text

Reviewer: This still needs revision for clarity. Make it clear that you are referring to the body lengths of individuals from which vertebral samples were collected.

Line 248: Same.

Answer: total length. Added in the text

Reviewer: See above

Line 278-282: If the sample was almost entirely composed of adults how was size at maturity determined? This needs to be clearly explanained in the discussion.

Line 305: Find a more accurate word than adjust. It does not work here or other laces it is used.

Answer: rewrited

Reviewer: Word “adjusted” is still used in several places throughout the manuscript.

Line 305: This is not true. It would depend on if all size classes are represented, variability in size at age, number of females and males samples, etc.

Answer: rewrited

Reviewer: Revise this sentence for correct grammar.

Line 315: What does “presents good calcification” mean?

Lines 317-318: What is meant by “visualization of the growth bands was adequate for the study, with 96.39% of the 317 vertebrae being used for reading, and consequent good visualization of the growth bands.”? This needs revision for clarity.

Answer: Revised and clarified

Reviewer: These revised sentences need further revision. For example, what is “it” in this sentence? Further, the degree of calcification is not the important point, which is vertebrae contain a reliable record of age that has been validated in a number of species and they are not thought to be reabsorbed to the degree some other calcified tissues are.

Lines 401-411: Should directly discuss some of the unlikely k values that were obtained and indicate that the growth model for males should be revisited based on this and other issues pointed out within the text. Also, speculate why there was an issue.

Answer: We better explained the sentence

Reviewer: How can this be the case when in almost all cases, elasmobranch females are larger than males and have lower growth constants. Also, use the word larger rather than higher in this context.

Line 428: Or that the growth model in the current study does not accurately describe growth. Are there other studies that show theoretical longevity is off based on longevity estimates from recaptured tagged sharks of known or approximately known age?

Answer: corrected

Reviewer: This revised sentence needs grammatical revision and clarification.

Round 3

Reviewer 1 Report

I have reviewed the revised manuscript and feel it is acceptable for publication from a scientific standpoint. However, grammatically, the paper has a number of issues that still need to be addressed (see first identified revision in proof and first sentence of the introduction for examples). At this point, I will leave this up to the editor and copy editor. Below are several comments that were not adequately addressed that the authors should attend to.

Line 155: This needs a more complete explanation. Why only adjust for six months when there is a range? Depending on how you adjust can significantly impact your growth model parameter estimates.

Answer: We elucidate. Anyway, given the biology of the species, we will invariably have a variation between the real time of formation of the first band for each individual and the estimated time.

Reviewer: This still needs clarification. I assume the intent is to convey that 0.5 was removed from all band counts to determine age.

Answer2: Indeed. The text needed clarification. We provided a better explanation

Reviewer: This is still not clear.

Lines 401-411: Should directly discuss some of the unlikely k values that were obtained and indicate that the growth model for males should be revisited based on this and other issues pointed out within the text. Also, speculate why there was an issue.

Answer: We better explained the sentence

Reviewer: How can this be the case when in almost all cases, elasmobranch females are larger than males and have lower growth constants. Also, use the word larger rather than higher in this context.

Answer2:We explained that this is an exception. We changed the word for larger

Reviewer: As this is counter to the trend for almost all other elasmobranchs, further explanation and potential sources of bias should be identified.

Line 431: Use of adjusted is incorrect. This sentence should be revised for clarity.

Line 440: Cite Cailliet properly. As written, it implies he did work on the same species as the current study.
